# Sequencing, annotation and analysis of the genome of *Oxycarenus hyalinipennis*, the cotton seed bug

Sam D Heraghty[1] , Sheina B Sim[2], Scott M Geib[2], Renee L Corpuz[2], Christina D Hoddle[3], Mark S Hoddle[3] , Colin S Brent[4], Dawn E Gundersen-Rindal[1], Michael E Sparks[1]

**The cotton seed bug, *Oxycarenus hyalinipennis* Costa (Hemiptera: Lygaeidae), is an invasive cotton pest detected in California, United States, in 2019. This report details the assembly and annotation of a reference genome for *O. hyalinipennis*, which will assist with the development of control strategies and provide evolutionary insight into this pest and hemipterans in general. The genome was sequenced from an adult male specimen using PacBio HiFi long reads and Element short-read sequencing of a HiC library. The assembly is 727,196,238 bp long with an N50 of 116 Mb. A total of 28,646 genes were identified by a customized annotation pipeline. Comparative genomics results suggest that chromosomal rearrangements may be common within Pentatomomorpha, based on syntenic analyses across all superfamilies. Comparison between available heteropteran genomes was used to identify changes in gene family size, as well as evolutionary differences, between taxa with different feeding strategies. Notably, herbivorous taxa had significantly larger genomes than predatory taxa. These genomic resources will be useful in efforts to understand and control this pest and advance the understanding of hemipteran evolution.**

## Introduction

The cotton seed bug, *O. hyalinipennis* Costa (Hemiptera: Lygaeidae), is a serious pest of cotton and many other crop plants in the order Malvales. Native to northern Africa, *O. hyalinipennis* has spread globally, and its range now includes Asia, Europe, South America and the Caribbean (Smith & Brambila, 2008). The cotton seed bug was recently detected in southern California, United States (U.S.) (Saveer et al, 2024). In 2010, this species was detected in the U.S. state of Florida. Eradication efforts were successful and the U.S. was considered cotton seed bug-free until the recent confirmation of populations in California (NAPPO, 2014; Saveer et al, 2024). Given its long history of successful

colonization and devastating impact on cotton, there is significant concern that *O. hyalinipennis* may become established in southern U.S. states (including Alabama, Georgia and Texas) where there is a major cotton growing industry. Given the economic importance of the U.S. cotton industry, worth more than $21 billion (USD) annually (Meyer & Dew, 2025), and the threat posed by *O. hyalinipennis*, there is considerable interest in developing effective control strategies for this pest. To this end, it is imperative to develop high-quality molecular reference resources to guide research efforts towards developing new control strategies and to obtain a greater understanding of the molecular biology of *O. hyalinipennis*.

This study presents the first de novo genome assembly and annotation for *O. hyalinipennis* and also uses a recent cotton seed bug transcriptome (Heraghty et al, 2026). To highlight the genome assembly's utility, it was used to anchor several comparative genomics analyses to better understand the biology of this species and to examine heteropteran evolution more broadly. Heteroptera, an infraorder within Hemiptera, which contains *O. hyalinipennis*, has a dynamic evolutionary history and contains a multitude of species of economic concern. As of July 2024, 933 heteropteran taxa were on the U.S. Regulated Plant Pest List (see https://www.aphis.usda.gov/plant-imports/regulated-pest-list). In addition to these agricultural pest species, there are also species of human concern (e.g., bed bugs in the genus *Cimex*), and beneficial insects used in biological control programs (e.g., predatory stinkbugs, including multiple species in the genus *Arma*). Aside from roles in agriculture and human health, Heteroptera is also a useful model system for addressing questions related to the evolution of different feeding strategies. Current evidence suggests that the ancestral heteropterans had a predatory lifestyle, but herbivory and hematophagy evolved independently multiple times (Song et al, 2024). Thus, study of the evolutionary trends within this group will be useful in advancing the understanding of the evolution of complex traits (such as hemipteran diet [Song et al, 2024; Wu et al, 2024]), as well as phenomena like convergent (e.g., environmental adaptation in mangroves [Xu et al, 2017; Cerca, 2023]) and parallel (e.g., a suite of

[1]USDA-ARS Invasive Insect Biocontrol and Behavior Laboratory, Beltsville, MD, USA   [2]USDA-ARS Daniel K. Inouye U.S. Pacific Basin Agricultural Research Center, Hilo, HI, USA   [3]Department of Entomology, University of California, Riverside, CA, USA   [4]USDA-ARS Arid-Land Agricultural Research Center, Maricopa, AZ, USA

Correspondence: michael.sparks2@usda.gov

different phenotypic traits in ants [Morandin et al, 2016; Cerca, 2023]) evolution.

In addition to generating inferences across Heteroptera, the *O. hyalinipennis* genome was used to look at evolutionary trends in three different gene families pertinent to invasive insect biology: carboxylesterases (COEs), cytochrome P450s (CYPs) and GSTs, which have been widely studied (Sparks et al, 2017, 2020, 2024; Volonté et al, 2022; Xu et al, 2023). These gene families are involved in xenobiotic detoxification, in which molecules are catabolized so that they can be harmlessly expelled (Cruse et al, 2023). They are potentially excellent targets for control efforts because evidence suggests that these gene families are involved in processes like insecticide resistance and overcoming plant chemical defenses (Volonté et al, 2022; Cruse et al, 2023).

In summary, this study had three major goals: (1) to produce a high-quality annotated reference genome for this pest species to facilitate further study into novel biocontrol approaches; (2) to use this genome to anchor comparative genomics analyses across Heteroptera with a particular focus on the evolution of different feeding lifestyles; and (3) to identify evolutionary patterns in specific gene families of interest based on previous research and their relevance to invasive insect biology.

# Results

## Assembly and annotation

The genome assembly of *O. hyalinipennis* had a total length of 727,196,238 bp, an overall N50 of 116,050,537 bp, and a GC content of 38.3% (see Table S1 and Fig S1). The assembly had a BUSCO score of 98.4% (97.2% of BUSCOs were complete and single, and 1.2% were complete and duplicated), with 0.9% of BUSCOs being fragmented and 0.7% being missing (i.e., totally absent) using the hemiptera_odb10 dataset (n = 2,510). Repeat annotation found that 67.39% of the genome was made up of repetitive content, with the majority consisting of unidentified elements (see Table S2). Of the repetitive content that could be identified, it primarily consisted of LTR elements (14.13%), DNA elements (4.65%) and LINEs (4.32%) (Table S2).

The protein-coding gene annotation pipeline, described in the Materials and Methods section below, generated 107,590 gene predictions overall, of which 106,914 were unique protein sequences and 107,007 unique coding sequences. A total of 75,783 gene models satisfied the minimum and maximum peptide length parameters of 100 and 6,000 aa, respectively, among which 75,208 protein and 75,288 coding sequences were unique. The set of length-bounded protein-coding genes was partitioned on the basis of sequence similarity to reference proteins in NCBI NR, Swiss-Prot and TREMBL as follows: gold = 4,227 (5.6%), silver = 12,468 (16.4%), bronze = 11,951 (15.8%) and non-podium = 47,137 (62.2%). Of the non-podium tranche, only 7,025 (14.9%) exhibited a hit in any of these protein databases. These genes failed to satisfy the criteria for any of the podium levels, and likely represent either false-positive gene calls, degraded pseudogenes, or novel sequences not yet captured by reference databases. Overall, the assembly

and podium-level annotations compare favorably with those of other hemipterans (see Table S1).

## Syntenic analysis across Pentatomomorpha

Examination of synteny maps between the cotton seed bug genome and representative chromosome-scale genomes provides several evolutionary insights. There is evidence for large-scale chromosomal rearrangements within Pentatomomorpha, with changes becoming more apparent at greater phylogenetic distances (see Fig 1A for phylogenetic relationships among the taxa being considered, Fig 1B for comparison of *O. hyalinipennis* against *Gonocerus acuteangulatus*, Fig 1C for comparison against *Nesidiocoris tenuis*, Fig 1D for comparison against *Pyrrhocoris apterus*, Fig 1E for comparison against *Aradus depressus* and Fig 1F for comparison against *Nezara viridula*). Using the USDA-ARS_CSB.1 chromosome of the cotton seed bug genome as an example, in all comparisons, this chromosome has genes that map to at least two, often more, other chromosomes in a given reference taxon (Fig 1). For instance, see the comparison against *P. apterus* genes from USDA-ARS_CSB.1 map to chromosomes CM078371.1–CM078375.1, and CM078377.1 (Fig 1D).

## Gene family evolution across Heteroptera

A total of 27,208 orthogroups were identified by OrthoFinder across all representative taxa. Of the identified orthogroups, 4,572 were annotated with at least one Gene Ontology (GO) term. There were 3,117 orthogroups found common to all taxa across the Pentatomomorpha and Cimicomorpha, regardless of whether they had a predacious or herbivorous feeding lifestyle (see Fig 2). This likely represents a core genomic toolkit found in all heteropterans. A total of 501 and 485 orthogroups were found in all Cimicomorpha and Pentatomomorpha taxa, respectively, regardless of their feeding strategies. A total of 137 and 189 orthogroups were found in all predatory and herbivorous taxa, respectively, regardless of lineage (Fig 2). In the orthogroups exclusive to predatory insects, several GO terms related to metal ion binding were observed, including GO:0005506—iron ion binding, GO:0005509—calcium ion binding, and GO:0008270—zinc ion binding. This may suggest that a core part of the predatory genomic toolkit is dedicated to metabolizing metal elements from dietary intake. In orthogroups exclusive to herbivorous insects, several GO terms involved in sensory functions were observed, such as GO:0005549—odorant binding, GO:0007608—sensory perception of smell, and GO:0019236—response to pheromone.

CAFE identified expansions and contractions across all heteropteran taxa surveyed (Fig 3), with the largest being seen in *Aelia acuminata* (n = 2,306) and in *Acanthosoma haemorrhoidale* (n = 2,157). The largest contractions were noted in *Rhynocoris fuscipes* (n = 1,629) and in *Ranatra chinensis* (n = 1,271). These trends are relatively similar only when looking at statistically significant expansions and contractions (see Table S3), although there are far fewer instances (e.g., 147 orthogroups had statistically significant expansions in *A. haemorrhoidale*). This suggests species with many small (i.e., insignificant) expansions or contractions are not less likely to have large (i.e., significant) changes in orthogroup size.

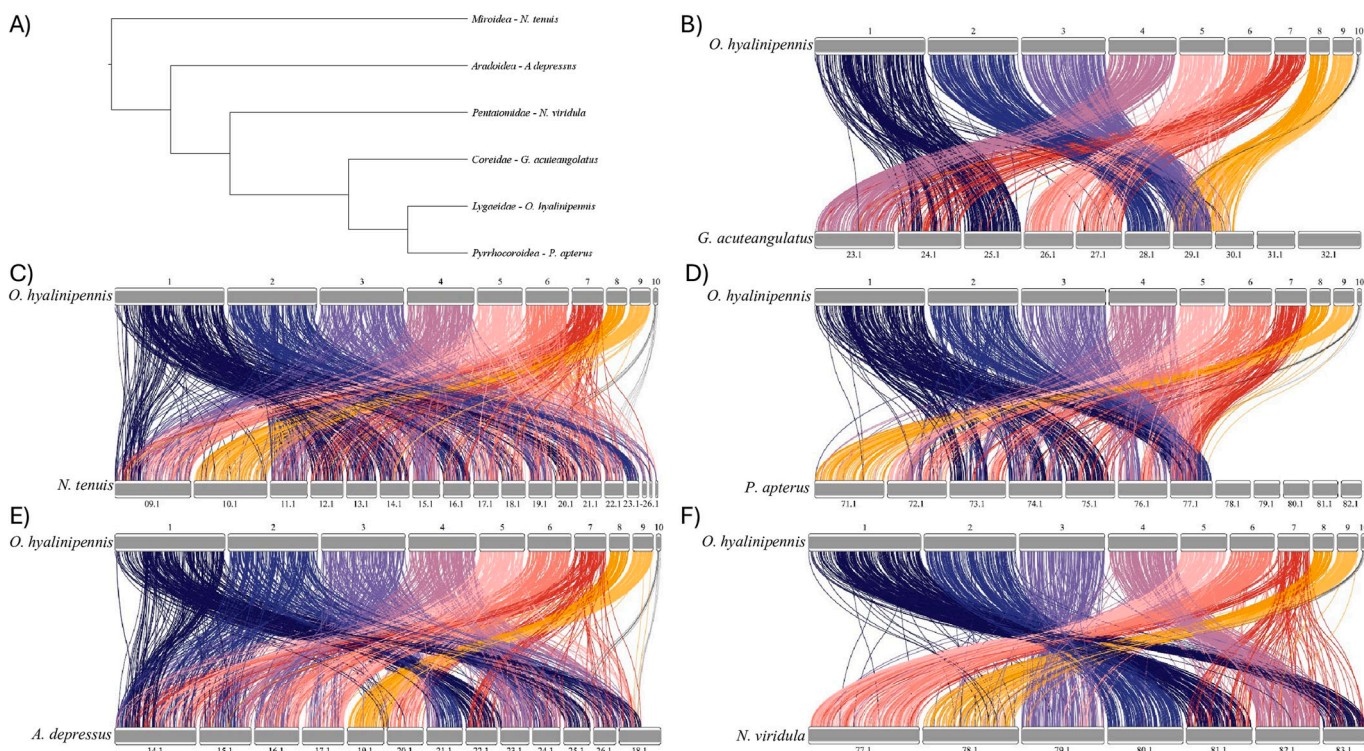

**Figure 1. Synteny of cotton seed bug across Pentatomomorpha.**
**(A)** A cladogram, distilled from (Johnson et al, 2018), illustrating the phylogenetic relationship between the superfamilies of Pentatomomorpha and a representative from its sister infraorder Cimicomorpha. **(B, C, D, E, F)** Syntenic maps of the location of each complete single-copy BUSCO gene found in *O. hyalinipennis* and the respective comparison taxa (*G. acuteangulatus*, *N. tenuis*, *Pyrrhocoris apterus*, *Aradus depressus*, and *N. viridula*). Note that for visualization purposes, chromosome names for comparator taxa have been shortened to remove redundant characters, specifically, for *G. acuteangulatus*, OX328123.1 becomes 23.1; for *N. tenuis*, AP028909.1 becomes 09.1; for *P. apterus*, CM078371.1 becomes 71.1; for *A. depressus*, chromosome OY759214.1 becomes 14.1, and for *N. viridula*, OV725077.1 becomes 77.1.

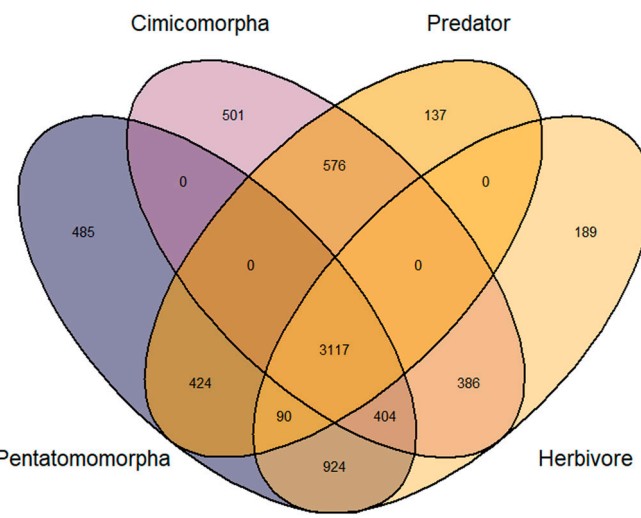

**Figure 2. Venn diagram showing the overlap of orthogroups found in all members of a given group.**
The "Predator" group contains orthogroups with at least one sequence from each taxon with a predatory lifestyle (see Table S3), the "Herbivore" group contains orthogroups with at least one sequence from each taxon with a herbivorous lifestyle, the "Pentatomomorpha" group contains orthogroups with at least one sequence from each taxon within the infraorder Pentatomomorpha and the "Cimicomorpha" group contains orthogroups with at least one sequence from each taxon within the infraorder Cimicomorpha.

Clustering of genes found in orthogroups with significant expansion was observed in all focal taxa (see Table 1). Most species had a nearly equal ratio of clustered to non-clustered genes. Although the average cluster size was consistent across taxa, ranging from 2.44 to 6.95 genes per cluster, there was a high degree of variation in the size of the largest gene cluster (Table 1). For instance, in some taxa, the largest cluster size is relatively small (e.g., n = 5 in *A. haemorrhoidale* and n = 5 in *Arma custos*) whereas in others, cluster sizes are much larger (e.g., n = 73 in *Pilophorus perplexus* and n = 44 in *O. hyalinipennis*).

Focusing on the cotton seed bug results, there were 14 GO terms significantly enriched in the significantly expanded dataset and eight GO terms significantly enriched in the significantly contracted dataset. All GO terms that were enriched in the expanded dataset were either related to membrane channel protein activities (e.g., GO:0016020—membrane, or GO:0015081—sodium ion transmembrane transporter activity) or sensory perception (e.g., GO:0007606—sensory perception of chemical stimulus, or GO:0004984—olfactory receptor activity), suggesting an important role of ion transport and sensory systems in the evolution of this species. The GO terms enriched in the contracted dataset were generally related to peptidase activity (e.g., GO:0004252—serine-type endopeptidase activity, or GO:0008233—peptidase activity). Analyses of gene clustering also identified two large gene clusters (n = 44 and n = 35) on the USDA-ARS_CSB.1 chromosome of the CSB

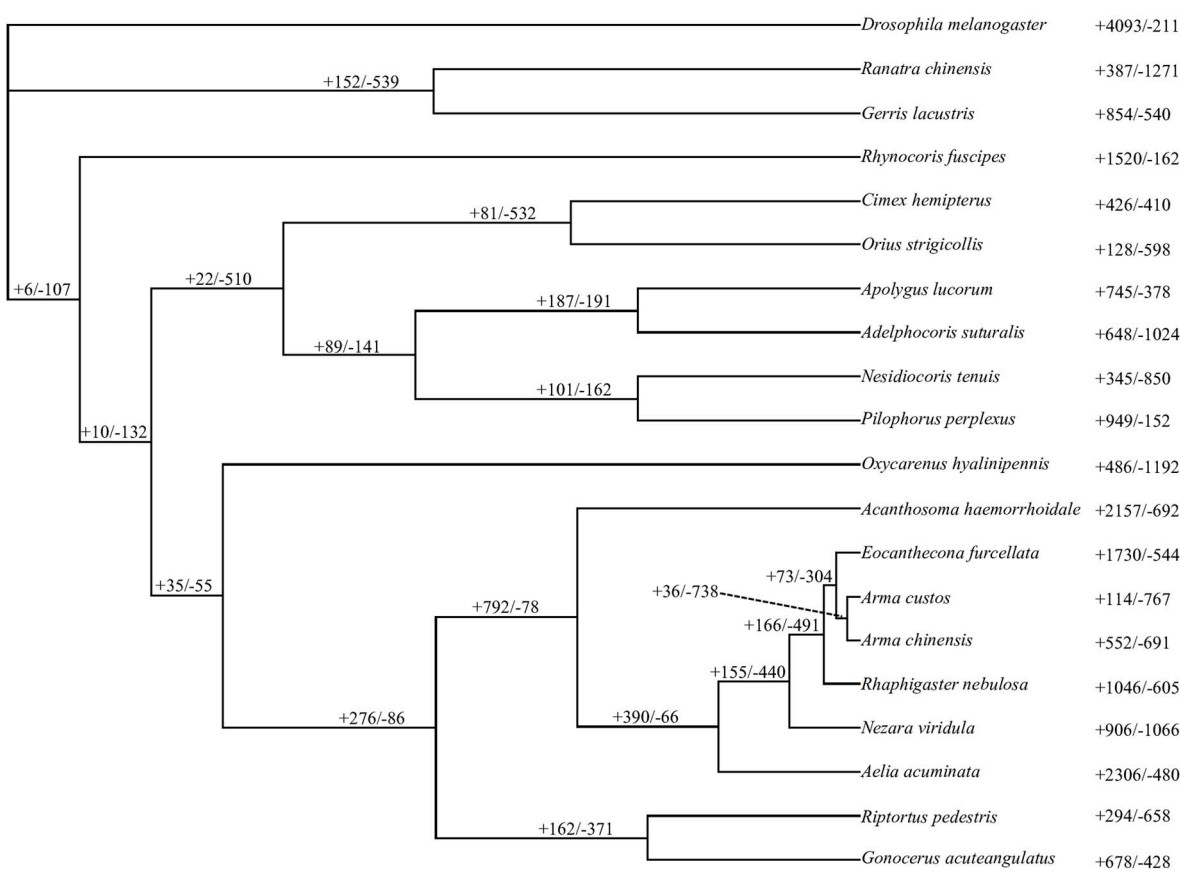

**Figure 3. CAFE results, showing the number of orthogroups that expanded (denoted with a "+") and contracted (denoted with a "−") for each taxon and internal node.**
Note: this shows all expansions and contractions, not only those that are statistically significant.

genome assembly, indicating a possible hotspot of evolution. The 44-gene cluster starts at chromosomal coordinate 24,954 and ends at 343,459, and the 35-gene cluster is contained in the interval of 2,228,369 through 2,445,624.

### Predacious versus herbivorous taxa

Of all metrics assessed (see the Materials and Methods section), only genome assembly size had a significant difference between herbivorous and predacious taxa ($t$ = −2.4957, $df$ = 14, $P$ = 0.02568), with the herbivorous taxa having larger genome assembly sizes (see Fig S2). In all other instances, the $P$-value was greater than 0.05 (results not shown). Both Blomberg's K (K = 0.358, $P$ = 0.043) and Pagel's lambda ($\lambda$ = 0.866, $P$ = 0.002) yield significant values, which suggests that a phylogenetic signal associated with genome size exists. The phylogenetic ANOVA detected a significant difference in genome size with predatory taxa having smaller genome sizes (F = 13.627491, $P$ = 0.0208). Similarly, the phylogenetic generalized least squares (PGLS) analysis produced a significant model ($R^2$ = 0. 1685, $P$ = 0.008515) with predatory taxa having a smaller genome size (estimate = −4.1 × $10^8$, $P$ = 0.008515). Finally, model comparison between the Brownian motion (Akaike information criterion [AIC] =

1710.552) and the Ornstein–Uhlenbeck (OU) process (AIC = 1708.464) found stronger support for the OU model (per the lower AIC value).

### Specific gene family evolution

#### Clustering analysis
From the podium-level gene set, a total of 35, 52 and 15 genes were recovered from the COE, CYP and GST gene families, respectively. Although gene clustering was observed in all three families, there was relatively little clustering in the COE and GST families (see Fig 4). However, when including the non-podium dataset, clustering considerably increased in the COE family, which better matches what is observed in other hemipterans (Volonté et al, 2022). The inclusion of non-podium genes dramatically increased the number of GSTs detected, from 15 to 86, which may suggest a dynamic evolutionary history in this gene family.

### Orthogroup analysis

There were 16, 34 and 6 orthogroups containing at least one cotton seed bug sequence for the COE, CYP and GST gene families, respectively (see the Table 2). In all cases, the largest number of orthogroups also contained at least one sequence from the brown

**Table 1.   Summary of the orthogroup expansion and spatial distribution with the following header titles: the number of significantly expanded orthogroups (#OGs), the number of genes located outside of gene clusters (# single), the number of genes within gene clusters (# clustered), # single ÷ # clustered (ratio), the largest individual cluster in each taxon (largest cluster size), and the average cluster size for a given taxon (average cluster size).**

| Species | # OGs | # Single | # Clustered | Ratio | Largest cluster size | Average cluster size |
|---|---|---|---|---|---|---|
| A. haemorrhoidale | 147 | 213 | 137 | 1.5547 | 5 | 2.45 |
| Adelphocoris suturalis | 135 | 425 | 1854 | 0.2292 | 23 | 3.83 |
| A. acuminata | 334 | 490 | 330 | 1.4848 | 12 | 2.64 |
| Apolygus lucorum | 166 | 334 | 454 | 0.7357 | 22 | 2.86 |
| A. chinensis | 220 | 653 | 6883 | 0.0949 | 51 | 6.01 |
| A. custos | 37 | 68 | 66 | 1.0303 | 5 | 2.44 |
| Cimex hemipterus | 41 | 115 | 129 | 0.8915 | 25 | 3.49 |
| Eocanthecona furcellata | 682 | 711 | 732 | 0.9713 | 24 | 3.14 |
| Gerris lacustris | 95 | 420 | 771 | 0.5447 | 15 | 3.31 |
| G. acuteangulatus | 129 | 209 | 378 | 0.5529 | 9 | 2.68 |
| N. tenuis | 61 | 30 | 136 | 0.2206 | 9 | 2.47 |
| N. viridula | 314 | 383 | 460 | 0.8326 | 29 | 3.24 |
| Orius strigicollis | 12 | 55 | 51 | 1.0784 | 7 | 2.83 |
| O. hyalinipennis | 38 | 30 | 278 | 0.1079 | 44 | 6.95 |
| P. perplexus | 239 | 353 | 752 | 0.4694 | 73 | 2.96 |
| R. chinensis | 20 | 28 | 72 | 0.3889 | 10 | 4.00 |
| Rhaphigaster nebulosa | 411 | 740 | 1425 | 0.5193 | 24 | 2.94 |
| R. fuscipes | 82 | 138 | 146 | 0.9452 | 26 | 3.11 |
| R. pedestris | 72 | 316 | 308 | 1.0260 | 16 | 2.99 |

marmorated stink bug, which is expected given that it is the most closely related species among the comparator taxa. Notably, nearly all the GST orthogroups included all species, suggesting a core shared hemipteran toolkit with little recent evolution. Similarly, a large portion of both the COE and CYP orthogroups contained at least one sequence from each species, suggesting the existence of a large, shared hemipteran xenobiotic detoxification toolkit. In these gene families, however, there is more recent evolutionary innovation as evidenced by the orthogroups consisting of only cotton seed bug sequences, or cotton seed bug and brown marmorated stink bug sequences.

# Discussion

## Spatial patterns in evolution

The work presented here yields several useful insights into evolutionary trends within Heteroptera that may help to spur further study and better guide the development of management strategies of both beneficial and pest species. The synteny analyses indicate that large chromosomal rearrangements are relatively common within this group. Given the commonness of such rearrangements, it is possible that structural variations (e.g., inversions or translocations) are also common within this group. As such, it may be

that structural variants are likely to be involved in adaptation to various phenomena. For instance, adaptive inversions capable of producing "supergenes" (Thompson & Jiggins, 2014) may be typical within this lineage. Identifying such inversions could provide a useful target for molecular control strategies for pest species in general. Alternatively, in beneficial species, adaptive inversions could be used to select populations that might be a better source from which to sample biological control agents. To verify if such structural variants exist will require large-scale population genomics studies of wild populations both in the native and invaded range. Such studies would also have the benefit of expanding the understanding of how invasions can alter population genomic patterns in general.

Clustering patterns were variable across both the cotton seed bug genome and the focal taxa used in the genome-wide expansion and contraction analyses. In both the COE and CYP gene families, roughly 40% of orthogroups have at least one sequence from each of the comparator taxa, suggesting that at least part of these gene families is relatively well conserved across the Hemiptera in general. In some cases, there has been more recent evolutionary innovation, which is consistent with expectations based on work in other taxa (Volonté et al, 2022). Within the cotton seed bug in particular, gene family analyses revealed evidence of clustering in both the CYP and COE families. The clustering patterns observed are most likely the result of tandem duplications, which

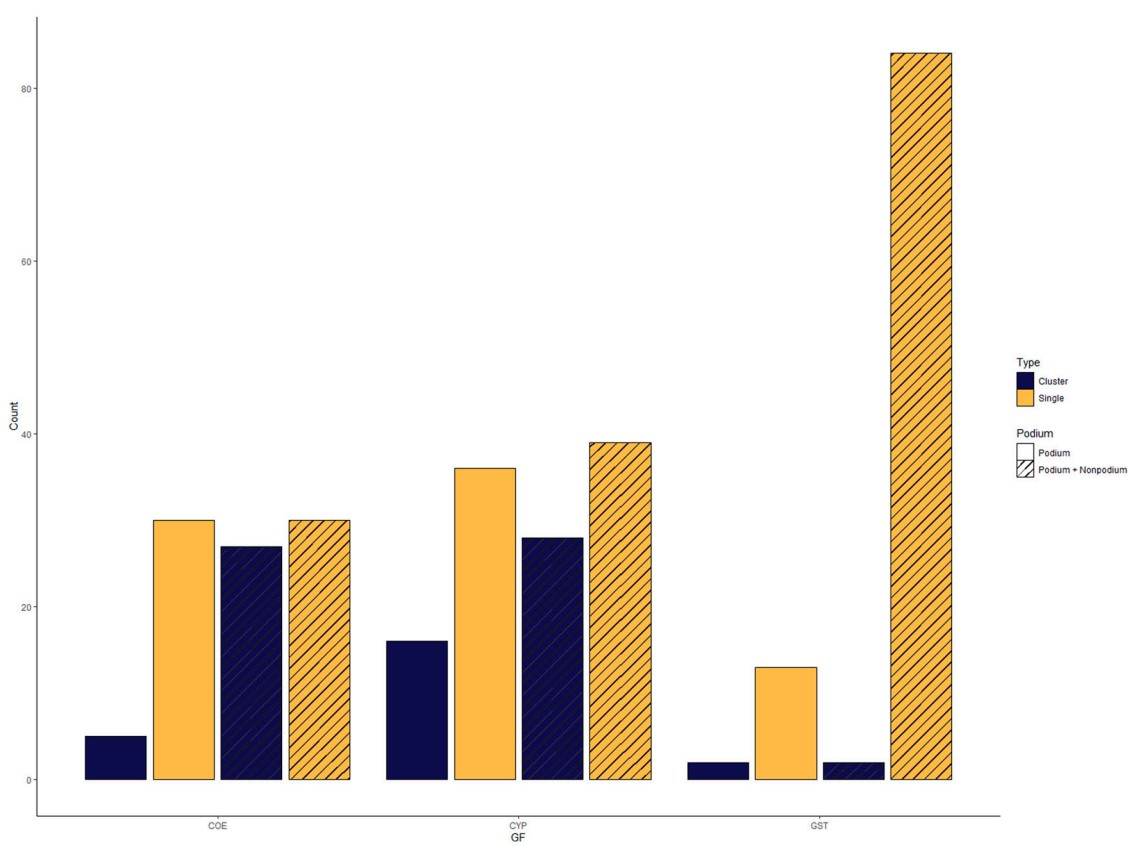

**Figure 4.   The number of genes in each gene family located in multi-gene clusters (i.e., genes within 35 kbp of at least one other gene within the same family) or if they were singletons (no genes of the same family within 35 kbp).**

**Table 2.   Summary table showing the distribution of orthogroups across different combinations of cotton seed bug and the comparator taxa with the following abbreviations: CSB ~ _O. hyalinipennis_, BMSB ~ _H. halys_, CLEC ~ _C. lectularius_ and DCIT ~ _D. citri_.**

|              | COE | CYP | GST |
|--------------|-----|-----|-----|
| CSB          | 2   | 1   | 0   |
| CSB+BMSB     | 4   | 8   | 0   |
| CSB+DCIT     | 1   | 2   | 0   |
| CSB+CLEC     | 0   | 2   | 0   |
| CSB+BMSB+CLEC| 3   | 6   | 1   |
| CSB+CLEC+DCIT| 0   | 1   | 0   |
| All          | 6   | 14  | 5   |
| Total        | 16  | 34  | 6   |

can result from unequal crossing over and ultimately produce multiple gene copies in close proximity (Lallemand et al, 2020).

There is some evidence suggesting that the mechanism of gene duplication might influence a given gene copy's evolutionary fate. Work in _Arabidopsis_ suggests that genes generated by tandem duplication might be more likely to undergo pseudogenization than those generated by other duplication mechanisms, although this might be lineage-specific (Hanada et al, 2008; Lallemand et al,

2020). Clustering in the COE gene family was not evident without the inclusion of COE candidates from the non-podium dataset, which may include pseudogenes, genes poorly represented in sequence databases or gene prediction errors. Therefore, it may be that tandem duplication is also more likely to result in pseudogenization in the Hemiptera as well as the Brassicales, suggestive of a substantially more fundamental biological process; however, far more extensive evolutionary analyses would be needed to test this hypothesis.

Interestingly, the GST family exhibits almost no clustering, even when the non-podium dataset is used, notwithstanding the fact that inclusion of GSTs from the non-podium dataset does dramatically increase the number of copies identified (see Fig 4). This suggests that, unlike the CYP and COE gene families, tandem duplication is unlikely to have a major role in the evolution of this gene family within the cotton seed bug. A more likely explanation for the pattern observed here is that copies of GST genes may have proliferated throughout the genome because of the action of transposable elements (TEs), which can duplicate genes either through retroposition or transduplication (Lallemand et al, 2020). TEs are also able to distribute gene copies widely throughout the genome, which means they may not always produce gene clusters, as is the more likely case with tandem duplications. Inserting gene copies away from requisite regulatory elements may result in the new gene copies becoming pseudogenized, which could explain

why there are so many GST copies observed among the non-podium gene prediction data. In addition, TEs may only transport partial copies of genes, which also could result in pseudogenes. Nearly all orthogroups had at least one GST sequence from each of the comparator taxa. This suggests that the GST toolkit may not have experienced recent evolution in cotton seed bug, which would result in more orthogroups with restricted distributions (e.g., species-specific or found only in other comparator taxa).

Gene clustering patterns were relatively similar across all the focal taxa surveyed in the CAFE analysis, with most having similar ratios of genes found inside and outside of clusters. This may suggest that all taxa considered have relatively similar likelihoods of experiencing orthogroup expansion from tandem duplications relative to other mechanisms (e.g., TEs or segmental duplications). However, there was notable variation in the maximum size of gene clusters, suggesting that certain taxa may have particular genomic regions that are especially prone to the kinds of replication slippage events that give rise to tandem duplications (Lallemand et al, 2020). These regions may merit further study to better understand the structural elements leading to such duplication events in heteropterans and to understand how such regions may influence the management of either pest or beneficial species. For instance, the cotton seed bug had two large gene clusters on the USDA-ARS_CSB.1 chromosome with many genes of unknown function. Depending on the role of these genes, they could represent either a target for molecular pest control efforts or provide cautionary information: In the former instance, some data suggest that more extensive repertoires of certain receptor genes could enhance the efficacy of pesticides (Cheng et al, 2025). Thus, molecular control strategies could be devised to increase the number of receptor genes in a pest species to increase insecticide susceptibility. In the latter, more adverse case, a greater number of genes with xenobiotic detoxification functions has long been known to be associated with the development of insecticide resistance (Devonshire & Sawicki, 1979). Therefore, in species known to have a large number of genes with such functions, managers may elect to modify how they deploy insecticides to minimize how quickly resistance might develop (e.g., raising the action threshold for insecticide applications or focusing on alternative pest management strategies such as biological control). These regions also might be hotspots of genomic differentiation at the population level, as they may be predisposed to copy number variation or newly copied genes contained therein may be undergoing intense selection leading to neofunctionalization; both factors might influence pest management decisions. For instance, it may be best to avoid designing molecular control techniques that target genes in hotspots because such genes may be more likely to "escape" the control technique via mutation or some other genetic change. To identify if this applies, targeted population genomics studies will need to be conducted (Dorant et al, 2020; Heraghty et al, 2023).

### Evolution in herbivores versus predators

The analyses presented in this study provide insight into the evolution of different feeding strategies within Heteroptera. The only difference observed between the predacious and herbivorous taxa using the annotated genome dataset was genome assembly length, with herbivorous taxa exhibiting larger genomes. The phylogenetic analyses also support this conclusion, as both the phylogenetically informed ANOVA and PGLS tests reported the same relationship. The phylogenetic analyses also provided evidence that there is statistically significant phylogenetic structuring associated with genome size and that an OU model best describes the evolution of this trait. The OU model is generally used to describe stabilizing selection (Harmon, 2019), in which selection pushes a trait towards an optimum value. Previous work has returned mixed evidence for the action of selection on genome size, with some theoretical work suggesting that genome size is neutral, whereas other studies suggest that large genomes can be maladaptive (Blommaert, 2020). However, simulation studies have found that when a species is far from an evolutionary optimum, genome size is greatly increased and subsequently decreases over time to a stable level as optimal fitness is achieved (Banse et al, 2024). Current evidence suggests that a predatory lifestyle is the ancestral state for Heteroptera, which implies that herbivory is a more recent evolutionary innovation within this lineage (Song et al, 2024). Therefore, it is plausible that the evolution of herbivory in Heteroptera resulted in a large increase in genome size if early herbivorous heteropterans were far from an optimum fitness level. The persistence of large genome sizes in herbivorous taxa may be because of insufficient time for stabilizing selection to reduce the genome size to an optimum value. Alternatively, herbivorous and predacious taxa may have different optimal values for genome size. The dataset used in this study was too limited in scope to employ more complex models, which can be useful in addressing such questions (Pienaar et al, 2026). A more expansive dataset could also enable evaluations of genome size in other feeding types that are observed in heteropteran insects, such as omnivore or hematophagous diets.

A key future direction based on these results will be to identify differences in genome composition between the predacious and herbivorous taxa. Although the results robustly indicated a difference in genome size, there was no observed difference in the number of genes or in the number of othrogroups that underwent expansion or contraction. One alternative mechanism that could explain the differences in genome size may be the amount of repeat content. Not all of the focal taxa have had their repeat content annotated to facilitate analyses to address this possibility, but both broad scale (Petersen et al, 2019; Cong et al, 2022) and more specific studies (for example, in Ensifera [Yuan et al, 2024], Acrididae [Zhao et al, 2025], Lampyridae [Lower et al, 2017] and the lepidopteran pest, *Lymantria dispar* [Hebert et al, 2019]) have found that repeat content drives differences in genome size in various insect lineages.

Another non-mutually exclusive explanation may be that herbivorous taxa could have large numbers of pseudogenes, which are sequences that, despite possibly appearing to be a functional gene, are transcriptionally silent and regarded as "evolutionary junk" (Troskie et al, 2021). However, despite being transcriptionally silent, pseudogenes can still play an important role in evolution as they can serve to be the raw material for new genes and have the potential ability to regulate the expression of fully functional genes (Troskie et al, 2021). It is possible that through either of these functions, pseudogenes may have had an important role in the

evolution of herbivory in Heteroptera, although a more targeted study will be required to assess this scenario. Given these relatively preliminary results on the evolution of genome size in Heteroptera, it may be that this group will prove to be a useful model system for understanding the forces that shape genome size evolution both in insects and more generally.

The orthogroups identified across the surveyed taxa provide the basis for a conserved toolkit that might underlie the evolution of both predacious and herbivorous life histories in insects. In the case of the predatory taxa, given that the ancestral heteropteran is believed to have had a predatory lifestyle (Song et al, 2024), this means any orthogroups found exclusively in all extant predatory taxa might represent a core toolkit that has persisted over relatively deep evolutionary time. For instance, the suite of GO terms related to metal binding in orthogroups occurring in all predatory taxa suggests that dealing with metals is a key challenge associated with a predatory lifestyle. Indeed, there is some evidence suggesting that bioaccumulation of metals may pose a challenge to predatory insects (Zhang et al, 2024). Herbivores may not face this challenge, which could explain why one or more of the herbivorous taxa surveyed have lost copies of genes from those orthogroups. Current evidence suggests herbivory may have evolved independently multiple times in Heteroptera (Song et al, 2024). If that is the case, any orthogroups present in all separately evolved herbivorous taxa might represent an instance of parallel evolution in which a given orthogroup was co-opted for an herbivorous lifestyle. For instance, several sensory orthogroups were exclusively found in all the herbivorous taxa surveyed; these could represent genes that were co-opted for processes such as finding host plants. Targeted laboratory studies will be needed to identify the functions of these genes and better assess if there is a specific function critical to an herbivorous lifestyle or if these genes were simply purged from at least one of the predatory insect genomes.

### Caveats

A key caveat to the results presented here, and many comparative genomics projects in general, is that not all genomes are annotated using the same workflow or supportive extrinsic evidence (e.g., mRNA evidence), which can potentially impact results. Using the two *Arma* species genomes as an example, despite their close phylogenetic relatedness, there is a large difference in the number of identified genes, with 20,853 being annotated in *Arma chinensis* (Fu et al, 2024) and 13,798 being annotated in *A. custos* (Wang et al, 2024). Despite this large difference in gene number, the total assembly size of both genomes is similar, which may suggest that the discrepancy in gene number can be attributed to differences in gene finding and annotation methodologies. Such discrepancies in turn might make it difficult for programs like CAFE to identify true instances of expansion and contraction. Future work would be useful in quantifying how different decisions in annotation methodology might influence these sorts of comparative genomic analyses.

It is also important to note that most of the orthogroups could not be annotated with a GO term, which can limit the insight gained from using the methods used in this report. For instance, although an expansion may be detected, without GO information, it is difficult to speculate on the functional ramifications of said expansion. This means future functional work will be needed to assess the functions of these unknown orthogroups, or that more in-depth computational modeling will be required. In addition, even when GO terms are available, more in-depth laboratory work is needed to specifically identify the functions of potential genes of interest.

### Conclusion

In summary, this study has produced the first high-quality reference genome for the cotton seed bug, which will be useful in directing new lines of inquiry from both pest management and evolutionary biology perspectives. The genome was used to anchor specific analyses aimed at understanding evolutionary patterns in the COE, CYP and GST gene families, given their relevance in invasive insect biology. Specifically, observation of variable clustering patterns across all three gene families suggests that different mechanisms may have worked to shape their evolutionary history. In addition, results supported chromosomal rearrangements being relatively common within Pentatomomorpha. Finally, broad phylogenetic analyses found differences in genome size between herbivorous and predacious lineages within Heteroptera, with herbivores having larger genome sizes than predators. Additional work will be needed to understand the evolutionary mechanisms that have led to these differences.

# Materials and Methods

### Genome assembly and annotation

High molecular weight DNA was extracted from the whole body of a single adult male *O. hyalinipennis* specimen using the QIAGEN MagAttract High molecular weight DNA Kit. DNA shearing was performed using the Diagenode Megaruptor 3 with the 20 kb fragment protocol. Sheared DNA was prepared for PacBio sequencing using the SMRTBell Express Template kit 2.0. The library was size-selected with beads to remove library molecules less than 3 kb in length to generate the final libraries for sequencing. Sequencing was performed on a Revio System using Binding Kit v2.0, Sequencing Kit v2.0, and Revio SMRT Cell. The library was sequenced using a 24-h movie time on one SMRTcell. Raw subreads were converted to HiFi data by processing with CCS to call a single high-quality consensus sequence for each molecule, using a 99.5% consensus accuracy cutoff.

Insect DNA was cross-linked, digested using the enzymes DdeI and DpnII, and proximity ligated. After proximity ligation and nucleic acid purification, DNA was size-selected to remove fragments smaller than ~250 bp. After fragmentation and sample purification using SPRI-beads, library preparation was performed using the NEB Next Ultra II DNA Library Prep Kit and followed by the Element Biosciences Adept Library Prep Kit. The resulting HiC library was sequenced on one partial flow cell of an Element Biosciences AVITI system. A de novo contig assembly was produced

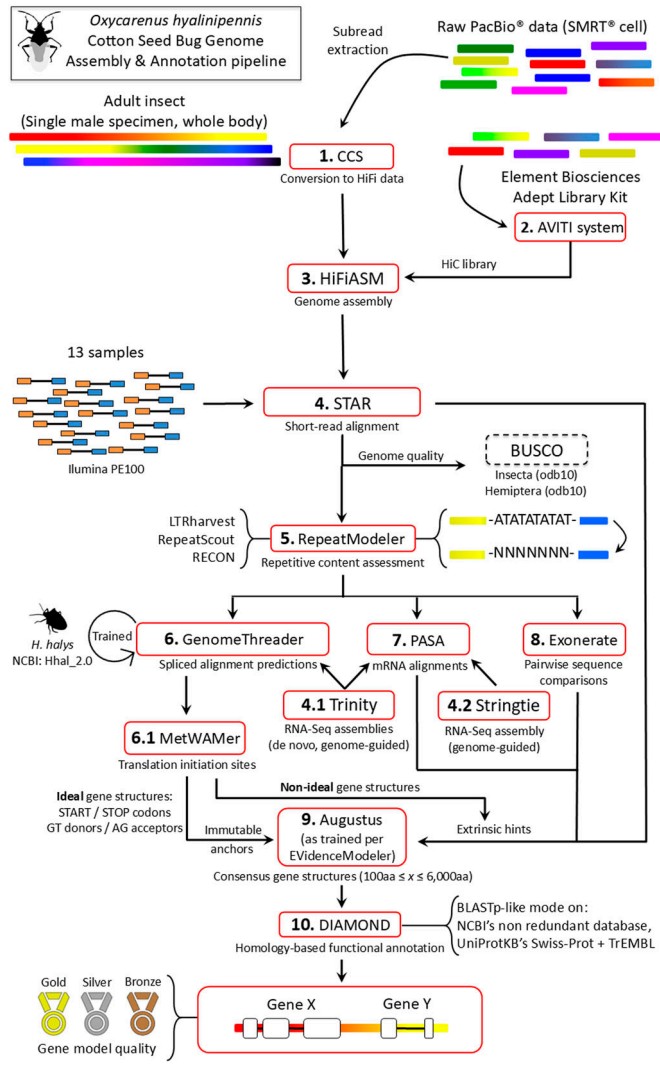

**Figure 5. *O. hyalinipennis* genome assembly and annotation pipeline.**
The automated pipeline used to assemble the cotton seed bug genome, as well as annotate repetitive content and nuclear protein-coding genes, is a modified version of the methods used for annotating genomes from the spongy moth (*Lymantria dispar* [Lepidoptera: Erebidae]) subspecies complex (Hebert et al, 2019; Sparks et al, 2021).

using HiFiASM v0.19.3-r572 (Cheng et al, 2021), and contigs were placed in a chromosomal context using HiC contact mapping.

To assess genome quality in terms of known single-copy orthologs, the chromosome-level assembly was compared with the hemiptera_odb10 database (Kriventseva et al, 2019) using BUSCO v5.7.1 (Simão et al, 2015). Repetitive content was identified using RepeatModeler (Smit & Hubley, 2015) and masked with RepeatMasker (Smit et al, 2015). Genome assembly quality was visualized by creating a snail plot via BlobTools v4.3.5 (Laetsch et al, 2017).

Gene annotation was conducted using a retrained version of an insect-specific pipeline previously used by the authors to annotate whole-genome assemblies of the spongy moth, *L. dispar* (Hebert et al, 2019; Sparks et al, 2021) (see Fig 5). High-quality cotton seed

bug genes among Exonerate (Slater & Birney, 2005) and PASA (Haas et al, 2003) results were identified by EVidenceModeler (Haas et al, 2008). These were used as training data for updating the internal models of Augustus (Stanke et al, 2006). Splice site models for GenomeThreader (Gremme et al, 2005) were retrained using BSSM4GSQ (Sparks & Brendel, 2005) with brown marmorated stink bug (*Halyomorpha halys*) data (Hhal_2.0, available from NCBI under accession GCF_000696795.2). Its predictions were post-processed with MetWAMer (https://github.com/scentiant/MetWAMer; Sparks & Brendel, 2008) to identify translation initiation sites. Ideal GenomeThreader-derived consensus gene structures comprised canonical translation start and stop codons and, for multi-exon genes, introns with GT-type donors and AG-type acceptors. Such ideal models were provided to Augustus as immutable anchors, effectively forcing them as-is into the results set, whereas other GenomeThreader-predicted models were supplied only as hints. RNA-Seq alignments to the cotton seed bug genome, as computed by the STAR aligner (Dobin et al, 2013), were also provided as extrinsic evidence to Augustus. These data were obtained from a publicly available cotton seed bug transcriptome (Heraghty et al, 2026). Both of the *alternatives-from-evidence* and *alternatives-from-sampling* Augustus flags were set to false. Protein sequences predicted by the overall pipeline were constrained to between 100 and 6,000 aa in length. These were then compared with the NCBI NR, and UniProtKB's Swiss-Prot and TrEMBL protein databases (UniProt Consortium, 2019) using the DIAMOND aligner in its BLASTp-like mode (Buchfink et al, 2015). Gene models were sorted into gold, silver and bronze tiers based on their encoded products' similarity to reference proteins. Gold-tier entries were at least 150 aa in length and exhibited a single high-scoring segment pair with a reference protein (also of at least 150 aa), such that 75% or more of aligned residues were positively similar, and the hit length-to-subject sequence length ratio was 90% or greater. Silver-tier entries were a minimum of 100 aa residues long with a hit spanning at least 75% of a reference protein sequence's length (also of at least 100 aa). Bronze-tier entries were similar, but with the hit covering 30% or more of the reference protein's length.

## Syntenic analyses

To evaluate large-scale patterns in chromosome evolution across the infraorder Pentatomomorpha, representative chromosome-level genomes from each super family were downloaded from NCBI, including *G. acuteangulatus*, *N. viridula*, *A. depressus* and *P. apterus* (see Table S4 for accession numbers and other details). In addition, a representative from the infraorder Cimicomorpha (*N. tenuis*) (Table S4), which is sister to Pentatomomorpha (Johnson et al, 2018), was included for comparison. For each of the representative genomes, BUSCO was used with the hemiptera_odb10 reference dataset to obtain genomic coordinates for all complete single-copy BUSCO genes within each genome. The cotton seed bug genome was then compared against each reference genome, and all genes that were complete and single-copy in both cotton seed bug and the given reference were retained and mapped against one another using RIdeogram

v0.2.2 (Hao et al, 2020). Resultant synteny maps were then manually analyzed.

## Gene evolution across Heteroptera

To facilitate evaluation of broader evolutionary trends within the suborder Heteroptera, representative genomes were searched for using the GoaT platform (accessed 24 February 2026 [Challis et al, 2023]) to identify all publicly available heteropteran genomes of at least the chromosome level with the following search: "tax_-tree(Heteroptera) AND assembly_level = chromosome"; in addition, it was required that these genomes were annotated (see Table S4 for a list of all genomes used for analysis). The *Drosophila melanogaster* (Release 6 [Hoskins et al, 2015]) genome was selected for use as an outgroup. OrthoFinder v2.4.1 (Emms & Kelly, 2019) was used to cluster annotated proteins of each representative taxa into orthogroups, which the software defines as groups of genes with a single shared evolutionary history. For each orthogroup, the longest sequence was identified and then annotated using InterProScan v5.71-102.0 (Jones et al, 2014). Orthogroups were grouped into one of four categories: present in all predacious taxa, present in all herbivorous taxa, present in all Pentatomomorpha, and present in all Cimicomorpha. The ggVenn v0.1.01 R package (Yan, 2023) was used to visualize overlaps between orthogroups found in the aforementioned groups. For groups of interest (e.g., orthogroups found in all predacious taxa), REVIGO v1.8.1 (Supek et al, 2011) was used to summarize associated GO terms.

All single-copy orthologs identified (n = 78) were multiply aligned using MUSCLE v3.8.1551 (Edgar, 2004). The resultant Fasta files were converted to PHYLIP format using the seqret program from EMBOSS v6.6.0.0 (Rice et al, 2000). Individual PHYLIP files were concatenated using the concat function from the AMAS Python module (Borowiec, 2016). The concatenated PHYLIP file was used to construct a phylogenetic tree using RAxML v1.2.1 (Stamatakis, 2014) in partitioned mode. Each partition corresponded to one of the single-copy orthologs, with all partitions using the JTT+G model. The resultant tree was converted into an ultrametric (time calibrated) tree (see Fig S3) using the chronos function in the ape R v5.8-1 package (Paradis & Schliep, 2019).

CAFE v5.1 software (Mendes et al, 2021) was used to identify orthogroups that have experienced either significant contractions or expansions in gene copy number over their evolutionary history ($P < 0.05$). After the recommended CAFE workflow, the provided *clade_and_size_filter.py* script was used to filter out any especially large orthogroups that might cause errors in parameter estimation. CAFE was then run in base mode and error corrected to account for mistakes in genome assembly, again after the recommended protocol.

For orthogroups shown to have experienced a significant contraction or expansion, a subsequent Gene Ontology (GO) analysis was performed. The GOFuncR v1.26.0 R package (Grote, 2024) was used to perform a GO enrichment analysis for each species. For this analysis, any orthogroup that had a significant change and was annotated with a GO term by InterProScan was coded as a "candidate," and all other orthogroups that were annotated with a GO term and contained at least one sequence from

the given taxa were listed as the background set. Orthogroups that experienced significant contraction and expansion were analyzed separately for each species.

To assess potential mechanisms driving gene expansion, genes that were members of orthogroups flagged as significantly expanded were identified as being clustered or non-clustered. A gene was considered clustered if it was within 35 kbp of another gene in an expanded orthogroup (regardless of whether or not they were in the same orthogroup). This definition of gene cluster was selected to be consistent with other comparative genomics work performed in hemipterans (Volonté et al, 2022). Clustering was performed using the *merge* function from bedtools v2.27.1 (Quinlan & Hall, 2010).

Several metrics were assessed to determine if there were significant differences between taxa with predatory and herbivorous feeding lifestyles: (1) ratio of clustered to non-clustered genes, (2) total number of expansions, (3) total number of significant expansions ($P < 0.05$), (4) total number of contractions, (5) total number of significant contractions ($P < 0.05$), (6) largest gene cluster, (7) average gene cluster, (8) genome assembly size, and (9) number of annotated genes. In instances where a species could not be assigned to one of these categories (e.g., for a species exhibiting a hematophagous or omnivorous lifestyle), it was excluded from analysis. In total, eight species were identified as predacious and eight as herbivorous (see Table S3 for details). For each metric, the *var.test* and *shapiro.test* functions in R were used (with default settings) to test that the dataset for each metric was normally distributed and that there was not a significant difference in variance. The *t.test* function was then used to determine if there indeed was a significant difference between their means (with the *var.equal* setting being set based on the output of the *var.test* function and the alternative setting being set based on data visualization using the *boxplot* function in R; default values were used for all other settings). In cases where a nonparametric test was required, the *wilcox.test* function was used.

To facilitate a more comprehensive phylogenetic analysis of diet type and genome evolution, all heteropteran genomes in the GoaT search, regardless of availability of genome annotation files, were used to investigate patterns in genome size as related to diet type (see Table S5 for all species used). The phylogenetic tree produced by the GoaT search was first pruned to only retain one representative genome per species, with the longest assembly being used for subsequent analysis. Branch lengths were estimated using the *compute.brlen* function from the ape R package (Paradis & Schliep, 2019) with the method argument set to "Grafen." The tree was then converted to an ultrametric tree (see Fig S4 for calibration points) using the *chronos* function from ape. The time-calibrated tree was subsequently pruned of any taxa that could not be reliably classified as either an herbivore or a predator. To test for a phylogenetic signal associated with genome size, both Blomberg's K and Pagel's lambda were calculated using the *phylosig* function from the phylotools v2.5-2 R package (Revell, 2024). A phylogenetically informed ANOVA was then conducted to identify if there was a significant difference in genome size between different diets when accounting for phylogeny using the *phylANOVA* function from the phylotools R package. In addition, a PGLS analysis was conducted to identify differences in assembly

size related to diet type when also accounting for phylogeny using the caper v1.0.4 R package's *pgls* function (Orme et al, 2025) with its lambda parameter set to "ml". Finally, to evaluate whether a Brownian motion or an Ornstein–Uhlenbeck (OU) process better explained the evolution of gene size with respect to diet, the *OUwie* function from the OUwie v3.01 R package was used (Beaulieu & O'meara, 2026). The phylogenetic tree was first coerced to be binary using the *multi2di* function from the ape R package. To model Brownian motion, the *OUwie* function was used with its model argument set to "BM1" and its algorithm argument set to "three.point." To model an OU process, the *OUwie* function was used with the model argument set to "OU1" and the algorithm argument set to "three.point." Models were then compared in terms of the AIC.

### Specific gene family evolution

Exemplar sequences of the COE, CYP, and GST gene families were sourced from a previous study (Sparks et al, 2024), comprising curated genes from hemipteran genome projects, including *Cimex lectularius* (GCF_000648675.2), *Diaphorina citri* (GCF_000475195.1) and *H. halys* (GCF_000696795.3). These reference sequences were used to identify corresponding gene family members in the podium-level datasets (i.e., gold, silver and bronze) of the newly annotated cotton seed bug genome using BLASTp v2.15 (Camacho et al, 2009). For each gene family, a bedfile was generated and then the bedtools v2.27.1 (Quinlan & Hall, 2010) *merge* function was used to identify any gene clusters, again using the definition mentioned above (Volonté et al, 2022). The newly identified cotton seed bug sequences and exemplar taxa were then organized into orthogroups using OrthoFinder v2.4.1 (Emms & Kelly, 2019) and subsequently filtered to retain only those that contained at least one cotton seed bug sequence.

## Data Availability

All data used in the preparation of the final genome assembly are available at the NCBI under BioProject accession number PRJNA1253754. PacBio long reads and Element short-read data are available at the SRA database under accession numbers SRX28484906 and SRX28484907, respectively, and the whole-genome assembly is available under GenBank accession number JBNFOR000000000. In addition, the whole-genome assembly and automated protein-coding gene finding results are available at the Open Science Framework repository at reference Sparks (2026). All publicly available data used in these analyses are listed in Tables S4 and S5. Lastly, interested parties are welcome to contact the corresponding author to request relevant data and software.

## Supplementary Information

## Acknowledgements

The authors thank two anonymous reviewers whose suggestions improved the quality of this report. Mention of trade names or commercial products in this publication is solely for the purpose of providing specific information and does not imply recommendation or endorsement by the U.S. Department of Agriculture. The USDA is an equal opportunity provider and employer.

### Author Contributions

SD Heraghty: conceptualization, data curation, formal analysis, investigation, visualization, methodology, and writing—original draft, review, and editing.
SB Sim: resources, formal analysis, methodology, and writing—original draft, review, and editing.
SM Geib: resources, formal analysis, methodology, and writing—original draft, review, and editing.
RL Corpuz: resources, formal analysis, methodology, and writing—original draft, review, and editing.
CD Hoddle: resources, and writing—original draft, review, and editing.
MS Hoddle: resources, and writing—original draft, review, and editing.
CS Brent: resources, and writing—original draft, review, and editing.
DE Gundersen-Rindal: conceptualization, resources, supervision, and writing—original draft, review, and editing.
ME Sparks: conceptualization, data curation, formal analysis, investigation, visualization, methodology, project administration, and writing—original draft, review, and editing.

### Conflict of Interest Statement

The authors declare that they have no conflict of interest.

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
