## [Reviewer comments · Life Science Alliance]

Sequencing, Annotation and Analysis of the Genome of *Oxycarenus hyalinipennis*, the Cotton Seed Bug

Sam Heraghty, Sheina Sim, Scott Geib, Renee Corpuz, Christina Hoddle, Mark Hoddle, Colin Brent, Dawn Gundersen-Rindal, and Michael Sparks

DOI: <https://doi.org/10.26508/lsa.202503598>

Corresponding author(s): Michael Sparks, Beltsville Agricultural Research Center

Review Timeline:

Submission Date:	2025-12-15
Editorial Decision:	2026-02-23
Revision Received:	2026-04-01
Editorial Decision:	2026-05-05
Revision Received:	2026-05-06
Accepted:	2026-05-07

Scientific Editor: Tim Fessenden

Transaction Report:

February 23, 2026

Re: Life Science Alliance manuscript #LSA-2025-03598-T

Michael E Sparks
USDA-ARS Beltsville Agricultural Research Center

Dear Dr. Sparks,

Thank you for submitting your manuscript entitled "Sequencing, Assembly and Annotation of the Whole-Insect Genome of *Oxycarenum hyalinipennis*, the Cotton Seed Bug" to Life Science Alliance. The manuscript was assessed by expert reviewers, whose comments are appended to this letter.

As you will see, both reviewers appreciate the importance and rigor of this sequencing effort as a resource to the community. While Reviewer 1 had no specific requests, Reviewer 2 made a few important points on which the study should be improved. We invite you to submit a revision to address their comments, in particular by foregrounding the important questions this study addresses and by examining genomic signatures of parasitic vs herbivorous lifestyles more closely.

To upload the revised version of your manuscript, please log in to your account: <https://lsa.msubmit.net/cgi-bin/main.plex>. You will be guided to complete the submission of your revised manuscript and to fill in all necessary information. Please get in touch in case you do not know or remember your login name.

B. MANUSCRIPT ORGANIZATION AND FORMATTING:

Thank you for this interesting contribution to Life Science Alliance. We are looking forward to receiving your revised manuscript.

Sincerely,

Tim Fessenden
Monitoring Editor
Life Science Alliance

Reviewer #1 (Comments to the Authors (Required)):

Summary

This paper, "Sequencing, Assembly and Annotation of the Whole-Insect Genome of *Oxycarenus hyalinipennis*, the Cotton Seed Bug", presents the first chromosome-level genome assembly and annotation of the cotton seed bug, a globally invasive and economically important cotton pest. The authors utilize multiple, high fidelity sequencing techniques to generate a high quality, highly contiguous genome with excellent completeness. They annotate protein-coding genes using a customized, evidence-rich pipeline which provided useful data to better understand the variation in genes particularly important in pest control development, and they leverage the genome to conduct comparative genomic analyses across Heteroptera, including gene family expansion/contraction, syntenic variation, and overall genome size variation. Overall, this research provides highly valuable information of ecological, evolutionary, and agricultural significance. More specifically, the generation of a high-quality genome and the utilization of the genome to investigate the evolutionary associations of genome size with feeding strategy and the gene family dynamics across the heteropterans offer advances to the field in terms of pest management and general zoological understanding of this diverse and ecologically important group of insects.

Main points

- A high-quality, chromosome-level genome assembly for *O. hyalinipennis* was generated

Yes, the data, including genome qc, completeness, and contiguity measurements are strongly supportive

- Large-scale chromosomal rearrangements are common within Pentatomomorpha

Yes, the data, including syntenic maps showing phylogenetically related rearrangements and multiple genes mapping to variable locations between *O. hyalinipennis* and other taxa are strongly supportive

- A conserved heteropteran genomic background exists alongside feeding-strategy-specific gene sets

Yes, the data, including specific GO enrichment based on specific orthogroups present in specific trophic lifestyles is strongly supportive.

- Herbivorous heteropterans have significantly larger genomes than predatory taxa

Yes, the data, including statistical analyses using high quality genome assemblies showing that repeats likely drive much of this change is strongly supportive

- Gene family evolution reveals lineage-specific expansion and clustering patterns

Yes, the data, including the differences in clustering of the detox-related gene families within *O. hyalinipennis*

I have no issues or concerns with this paper.

Reviewer #2 (Comments to the Authors (Required)):

Review of "Sequencing, Assembly and Annotation of the Whole-Insect Genome of *Oxycarenus hyalinipennis*, the Cotton Seed Bug" by Heraghty et al.

This article describes the genome of the Cotton Seed Bug and evaluates genomic patterns within the context of phylogenetic relatedness and herbivorous and predatory lifestyles. Overall I find the work is rigorous, thorough and a substantial contribution to genome resources in this group. There are a few areas where the article could be improved a few areas that could be interesting for formal inquiry.

The data presentation and analyses are of high-quality but the manuscript could be improved with more defined articulation and sequencing of specific scientific questions and the data that answer / address them. The manuscript is presented as a data generation resource and the scientific questions are somewhat briefly described and investigated. I found those to be the most compelling parts of the manuscript and if those were better developed I think the work would reach a much broader audience.

For example, one core question is are there defined genomic signatures that accompany the evolution of parasitic or herbivorous lifestyles? Here the authors find some evidence but both the question, data and analyses are diffusely spread throughout the manuscript. Similarly, larger questions of chromosome and molecular evolution are interesting but are presented as secondary to their core interest of generating agricultural resources.

The data and analyses presented suggest that there actually could be defined genomic signatures but given the current data the authors don't have the resolution to identify. A larger dataset with informed statistically rigorous phylogenetic comparative

analyses could provide a lot of insight into the herbivore / parasitic lifestyle questions. It's not clear to me how much data is available in this group but a broader analysis across a wider phylogenetic scale could be a very compelling, high-impact work. A recent review here describes phylogenetic comparative methods, for example data types and sample size limitations <https://doi.org/10.1093/jeb/voaf113>

Michael E. Sparks, Ph.D.
USDA-ARS Invasive Insect Biocontrol and Behavior Laboratory
Beltsville, Maryland, USA

April 1, 2026

Dear Dr. Tim Fessenden (and additional *Life Science Alliance* editorial staff),

We are pleased to submit for your consideration a revised manuscript describing the sequencing, assembly, annotation and analysis of the genome of the cotton seed bug, *Oxycarenus hyalinipennis*. We are especially grateful to the reviewers for their constructive comments and hope that the work will now be considered suitable for publication.

We have copied reviewer feedback below and indicated our responses using red font. In addition to the specific changes made in response to reviewer requests, we have also made various edits to the manuscript to comply with formatting requirements of *Life Science Alliance*, as well as sundry changes to improve the manuscript's readability.

Please do not hesitate to contact me at michael.sparks2@usda.gov if you have any questions or concerns.

Sincerely,

Michael E. Sparks and co-authors, 4-1-26

Reviewer #1 (Comments to the Authors (Required)):

Summary

This paper, "Sequencing, Assembly and Annotation of the Whole-Insect Genome of Oxycarenus hyalinipennis, the Cotton Seed Bug", presents the first chromosome-level genome assembly and annotation of the cotton seed bug, a globally invasive and economically important cotton pest. The authors utilize multiple, high fidelity sequencing techniques to generate a high quality, highly contiguous genome with excellent completeness. They annotate protein-coding genes using a customized, evidence-rich pipeline which provided useful data to better understand the variation in genes particularly important in pest control development, and they leverage the genome to conduct comparative genomic analyses across Heteroptera, including gene family expansion/contraction, syntenic variation, and overall genome size variation. Overall, this research provides highly valuable information of ecological, evolutionary, and agricultural significance. More specifically, the generation of a high-quality genome and the utilization of the genome to investigate the evolutionary associations of genome size with feeding strategy and the gene family dynamics across the heteropterans offer advances to the field in terms of pest management and general zoological understanding of this diverse and ecologically important group of insects.

Main points

• *A high-quality, chromosome-level genome assembly for *O. hyalinipennis* was generated*
Yes, the data, including genome qc, completeness, and contiguity measurements are strongly supportive

• *Large-scale chromosomal rearrangements are common within Pentatomomorpha*
*Yes, the data, including syntenic maps showing phylogenetically related rearrangements and multiple genes mapping to variable locations between *O. hyalinipennis* and other taxa are strongly supportive*

• *A conserved heteropteran genomic background exists alongside feeding-strategy-specific gene sets*
Yes, the data, including specific GO enrichment based on specific orthogroups present in specific trophic lifestyles is strongly supportive.

• *Herbivorous heteropterans have significantly larger genomes than predatory taxa*
Yes, the data, including statistical analyses using high quality genome assemblies showing that repeats likely drive much of this change is strongly supportive

• *Gene family evolution reveals lineage-specific expansion and clustering patterns*
*Yes, the data, including the differences in clustering of the detox-related gene families within *O. hyalinipennis**

I have no issues or concerns with this paper.

Response: **We thank Reviewer #1 for their time and kind comments.**

Reviewer #2 (Comments to the Authors (Required)):

*Review of "Sequencing, Assembly and Annotation of the Whole-Insect Genome of *Oxycarenus hyalinipennis*, the Cotton Seed Bug" by Heraghty et al.*

This article describes the genome of the Cotton Seed Bug and evaluates genomic patterns within the context of phylogenetic relatedness and herbivorous and predatory lifestyles. Overall I find the work is rigorous, thorough and a substantial contribution to genome resources in this group. There are a few areas where the article could be improved a few areas that could be interesting for formal inquiry.

The data presentation and analyses are of high-quality but the manuscript could be improved with more defined articulation and sequencing of specific scientific questions and the data that answer / address them. The manuscript is presented as a data generation resource and the scientific questions are somewhat briefly described and investigated. I found those to be the most compelling parts of the manuscript and if those were better developed I think the work would reach a much broader audience.

Response: We thank the reviewer for pointing out this issue with the presentation of the manuscript. Although a key motivation for this study was to make available these genomic resources (which will be useful in dealing with this prominent and destructive invasive pest in a timely manner), we nonetheless agree that it is important to present the scientific questions addressed in a manner that readers can more easily digest. To this end, we have added text to the Introduction section that clearly outlines the three principle research goals being addressed in this work (see lines 76-80). We have also expanded the dataset used and have performed additional comparative analyses as suggested by the reviewer (please see our remarks below). Finally, we have expanded the Discussion section to both accommodate the results of the newly included analyses, as well as to better highlight the evolutionary conclusions that may be drawn from our results (see lines 269-311 and 351-360).

For example, one core question is are there defined genomic signatures that accompany the evolution of parasitic or herbivorous lifestyles? Here the authors find some evidence but both the question, data and analyses are diffusely spread throughout the manuscript. Similarly, larger questions of chromosome and molecular evolution are interesting but are presented as secondary to their core interest of generating agricultural resources.

The data and analyses presented suggest that there actually could be defined genomic signatures but given the current data the authors don't have the resolution to identify. A larger dataset with informed statistically rigorous phylogenetic comparative analyses could provide a lot of insight into the herbivore / parasitic lifestyle questions. It's not clear to me how much data is available in this group but a broader analysis across a wider phylogenetic scale could be a very compelling, high-impact work. A recent review here describes phylogenetic comparative methods, for example data types and sample size limitations <https://doi.org/10.1093/jeb/voaf113>

Response:

We thank Reviewer #2 for the comment and for sharing the article. We do agree that a larger dataset and more rigorous comparative analysis could yield a more interesting result. However, there are some key limitations in terms of data availability in the biological domain we are operating in: within our focal lineage (Heteroptera), there are relatively few genomes that are assembled to chromosome scale and that also have publicly available, high-quality annotations. Dealing with the data gaps that would arise in analyzing the entire order would require that we generate

additional genomic resources, which unfortunately is beyond the scope of this study and is also not feasible with the research dollars we currently have available (either in amount or in the approved use of such funds as per the U.S. Congress). It is our hope in the future that such analyses will be possible as genomic resources are continually produced by the entomological community.

Despite these data limitations, we nevertheless were able to augment our dataset with two additional species of predatory insects whose genomes were published in the interim period between completing our initial analyses and receiving peer reviews. The incorporation of these two additional species does not substantively change our results. Further, we have extended analyses of the relationship between diet and genome assembly based on the reviewer's very constructive suggestion of using a more phylogenetically robust approach. For these analyses, we were able to greatly augment our dataset by including 20 additional species that had chromosome-length genome assemblies (but were not annotated). (As an aside: we did attempt to source annotations for these species by contacting some of the organizations producing genome assemblies, including the Darwin Tree of Life, but were informed that there was a significant backlog and very little likelihood that our various species of interest could "jump the queue.")

Please see 483-505 in the Materials and Methods section and 160-167 in Results for a more detailed description of the additional analyses we have performed as per the reviewer's suggestions. A brief summary of these contributions is as follows: Using our expanded dataset, we sought to determine whether there was phylogenic structuring related to the evolution of genome size across our dataset by calculating Blomberg's K and Pagel's λ (lambda). We then wanted to stringently assess if there was a difference in genome size between the two different diet types via a phylogenetically-informed ANOVA and also by a phylogenetic generalized least squares (PGLS) analysis. Finally, to investigate the mechanism of evolution governing genome size as it relates to diet type, we looked at both a Brownian Motion and Ornstein-Uhlenbeck (OU) model in a manner similar to that performed by Pienaar et al. 2026, although our analyses are much less complex than were those. Indeed, when attempting to fit more complex models (e.g., the OUwie program's "OUM" model as opposed to the "OU1" model used in our study), we received error messages from the software alerting us that there were not enough data available.

May 5, 2026

RE: Life Science Alliance Manuscript #LSA-2025-03598-TR

Dr. Michael E Sparks
Beltsville Agricultural Research Center
Invasive Insect Biocontrol and Behavior Laboratory
Bldg 004, Rm 016, BARC-West
10300 Baltimore Ave
Beltsville, MD 20705

Dear Dr. Sparks,

Thank you for submitting your revised manuscript entitled "Sequencing, Annotation and Analysis of the Genome of *Oxycarenus hyalinipennis*, the Cotton Seed Bug". We returned this to Reviewer 2 and as you can see below this referee is satisfied with no further requests. We would be happy to publish your paper in Life Science Alliance pending final revisions necessary to meet our formatting guidelines.

MANUSCRIPT ORGANIZATION AND FORMATTING:

To avoid unnecessary delays in the acceptance and publication of your paper, please read the following information carefully. Full guidelines are available on our Instructions for Authors page, <https://www.life-science-alliance.org/authors>

- Please add the X and Bluesky handles of your host institute/organization, as well as your own, and/or one of the authors, in our system.
- Figure 4 has only one panel; therefore, please remove the label A from the current figure.
- Please remove the title and list of the authors from page 34.
- Please add callouts for Figure 1A-C; and E, F to your main manuscript text.

We welcome submissions of potential cover images for the issue of LSA in which your work would appear. If you have high quality images associated with this work, please feel free to email these, with a caption, to the journal office.

LSA encourages authors to provide a 30-60 second video where the study is briefly explained. These videos will be appear embedded with the manuscript online at Life Science Alliance, and on social media to promote the published paper and authors (for examples, see <https://docs.google.com/document/d/1-UWCfbE4pGcDdcgzcmiuJl2XMBJnxKYeqRvLLrLSo8s/edit?usp=sharing>). Corresponding or first-authors are welcome to submit the video. Please submit only one video per manuscript. The video can be emailed to contact@life-science-alliance.org

FINAL FILES:

The following items are required for acceptance.

- An editable version of the final text (.DOC or .DOCX) is needed for copyediting (no PDFs).

The license to publish form must be signed before your manuscript can be sent to production. A link to the license to publish form will be available to the corresponding author only. Please take a moment to check your funder requirements.

Thank you for your attention to these final processing requirements. Please revise and format the manuscript and upload materials as soon as you are able.

Thank you for this interesting contribution to the literature. We look forward to publishing your paper in Life Science Alliance.

Sincerely,

Reviewer #2 (Comments to the Authors (Required)):

The authors have done a great job in revising the manuscript and I think it is a solid contribution to both resources and literature.

May 7, 2026

RE: Life Science Alliance Manuscript #LSA-2025-03598-TRR

Dr. Michael E Sparks
Beltsville Agricultural Research Center
Invasive Insect Biocontrol and Behavior Laboratory
Bldg 004, Rm 016, BARC-West
10300 Baltimore Ave
Beltsville, MD 20705

Dear Dr. Sparks,

Thank you for submitting your Research Article entitled "Sequencing, Annotation and Analysis of the Genome of *Oxycarenus hyalinipennis*, the Cotton Seed Bug". It is a pleasure to let you know that your manuscript is now accepted for publication in Life Science Alliance. Congratulations on this interesting work.

Your article will publish open access upon publication under a CC-BY license.

DISTRIBUTION OF MATERIALS:

Again, congratulations on a very nice paper. I hope you found the review process to be constructive and are pleased with how the manuscript was handled editorially. We look forward to future exciting submissions from your lab.

Sincerely,
